# Diversifying Language Models for
# Lesser-studied Languages and Language-usage Contexts:
# A Case of Second Language Korean

**Hakyung Sung**
Department of Linguistics
University of Oregon
hsung@uoregon.edu

**Gyu-Ho Shin**
Department of Linguistics
University of Illinois at Chicago
ghshin@uic.edu

## Abstract

This study investigates the extent to which currently available morpheme parsers/taggers apply to lesser-studied languages and language-usage contexts, specifically focusing on second language (L2) Korean. We pursue this inquiry by (1) training a neural-network model (pre-trained on first language [L1] Korean data) on varying L2 datasets and (2) measuring its morpheme parsing/tagging performance on L2 test sets from both the same and different sources of L2 training sets. The results show that the L2 trained models generally excel in L2 domain-specific parsing and tagging tasks compared to the L1 pre-trained baseline model. Interestingly, increasing the size of L2 training data does not lead to improving model performance consistently.

## 1 Introduction

Computational accounts for language science are gaining momentum in bringing together theories and data on second language (L2) research, offering promising directions toward interdisciplinary collaboration (e.g., Chapelle, 2001; Kyle, 2021; Meurers and Dickinson, 2017; Thewissen, 2013). This often involves leveraging NLP techniques to address learner language characteristics in sizable L2 corpora. Specifically, automatic processing of L2 data, including part-of-speech (POS) tagging, has become crucial to better understand the characteristics of learner language (e.g., Bestgen and Granger, 2014; Biber et al., 2016; Granger and Bestgen, 2017).

Despite the emerging trend, we identify two major concerns in the field. One lies in the research practice that applies the currently available NLP tools, predominantly trained on first language (L1) and general-purpose data, directly to L2 data. Considering the fact that no single corpus perfectly captures all aspects of how language is used (Meurers and Dickinson, 2017), it is reasonable to think that the types and characteristics of language-usage data affect the way in which specific inquiries are investigated properly on the basis of those data (Biber, 1993). Given this aspect, it may be the case that currently available parsing/tagging models, exclusively trained and evaluated on L1 data, do not work reliably and optimally for L2 data (cf. Kyle et al., 2022; Lan et al., 2023; Sung and Shin, 2023).

Another concern is the commitment to Diversity, Equity, and Inclusion (DEI) within research practice. Notably, there is a strong bias to a limited range of languages (e.g., [L2-]English) and language-usage contexts (e.g., general-purpose L1 usage) (Bender et al., 2021). This sampling bias poses a threat to DEI in the field, as well as weakening the generalizability of implications from previous findings to wider situations. This calls for researchers' attention to lesser-studied languages and language-usage contexts.

With this background, we investigate how well the current morpheme parsers/taggers cater to L2 Korean, a lesser-studied language and language-usage context. In particular, we explore a potential boost in performance when an L1 pre-trained model is additionally trained on L2 data with varying sizes, annotation schemes, and data cleaning processes. To preview, the results demonstrate that (1) models fine-tuned with L2 data outperform the L1 pre-trained baseline when evaluated on test sets from the same domain and (2) simply increasing the size of the L2 training data does not invariably lead to improved the fine-tuned models' performance.

This paper is structured as follows: We discuss the importance of morphological analysis in Korean studies and how morphological analyzers are applied in L2-Korean research (§2). Next, we outline experiments including datasets and evaluation metrics (§3). We delve into two subsequent experiments: the first involves fine-tuning an L1 model using the full scope of the L2 datasets (§4), while

the second focuses on rigorous data cleaning in one of the L2 datasets (§5). Comprehensive analyses of morpheme tokenization and POS tagging accuracy for both experiments are presented. Finally, we summarize our findings and propose future directions (§6).

## 2 Background

### 2.1 Morphological analysis of Korean

Korean has unique features such as its agglutinative nature and a Subject–Object–Verb word order (but relatively flexible due to overt case-marking via grammatical markers and context-dependent operation). Additional complexity is added by the use of speech levels (e.g., honorifics reflecting formality) and particles/affixes that sometimes lead to formal changes of root words to which they are attached (Sohn, 1999).

At the heart of these complexities involving Korean is the identification of a morpheme—the smallest meaningful unit in a language. Specifically, a word (dubbed an *eojeol* in Korean) often comprises a combination of several morphemes, each of which carries distinctive meanings and/or functions. To illustrate, while 책 *chayk* "book" consists of one single morpheme, 책장 *chaykcang* "bookshelf" involves two different morphemes, each of which has distinctive meaning: 책 *chayk* "book"+장 *cang* "shelf, case". A more complex example can be found in a predicate. Take 보였다 *poyessta* "was seen" as an example. This verb comprises a verb root 보- *po-* "to see", followed by (1) a passive-voice suffix -이- *-i-*, (2) a past-tense marker -었- *-ess-*, and (3) a declarative ending -다 *-ta*. On top of this combination, it involves re-syllabification of the passive-voice suffix and the past-tense marker, resulting in formal change of the given morphemes (i.e. *-i-* + *-ess-* → *-yess-*).

Given the intricacy of its morpheme composition, automatic analysis of Korean language hinges crucially upon accurate and appropriate identification of morphemes. Indeed, previous studies have highlighted the importance of morpheme-level parsing and tagging of Korean, implementing various approaches (e.g., morpho-syntactic rule-based approach, probabilistic method, grapheme-level approach, conditional random field method) to handle the composition of Korean morphemes (e.g., Choi et al., 2016; Han and Palmer, 2004; Lee and Rim, 2005, 2009; Na, 2015).

### 2.2 Application of morphological analyzers in L2-Korean research

Research on L2 Korean is increasingly leveraging automatic morphological analyzers (i.e., parsers/taggers). This includes, but is not limited to, lexico-grammatical feature analysis, automatic writing evaluation, and text analysis.

**Lexico-grammatical feature analysis**: In early studies, researchers investigated Korean particles as a core lexico-grammatical feature using an automatic morpheme parser/tagger. For example, Nam and Hong (2014) investigated L2-Korean learners' spoken production by comparing their use of particles across proficiency levels using a POS tagger (*Utagger*). Similarly, Dickinson et al. (2010) and Lee et al. (2016) incorporated an automatic POS tagger with a proposed automatic error-detection scheme to detect L2-Korean learners' non-prototypical use of particles. Building on this, Kim et al. (2016) explored the grammatical patterns in L2-Korean learners' written productions. They tagged morphemes in the target corpus and examined the resulting grammatical structures. Meanwhile, Jung (2022) and Shin and Jung (2022) analyzed the distribution of particle use in L2-Korean textbooks, by creating a pipeline that automatically extracts (using a *UDpipe* tagger) the target particles.

**Automatic writing evaluation**: Lim et al. (2022) developed an automated system for evaluating L2-Korean writing. This system utilized a transformer-based multilingual model alongside XLM-RoBERTa. Considering morphemes as one of the important measurable complexity features in the production data, an automatic POS tagger was employed. The developed tool was then applied to evaluate the writing proficiency of the L2-Korean learners. More recently, Hwang (2023) proposed a linguistic analysis tool that measures morphological complexity of L2-Korean production data based on an automatic morpheme tagger (*Kkma*). By automatically parsing morpheme and calculating L2 assessment indices, this study suggested its potential to model L2 Korean development.

**Text analysis**: Cho and Park (2018) applied four distinct automatic morphological analyzers (*Kkma, Okt, Hannanum, and Komoran*) to 16

essays penned by L2-Korean learners. To evaluate the text similarity of these essays, they utilized the TF-IDF (Term Frequency-Inverse Document Frequency) method, incorporating extracted morphemes as a key feature.

## 2.3 Evaluation of morphological analyzers on L2-Korean data

A recent finding points to a considerable gap in performance of currently available Korean morphological analyzers when applied to L2-Korean texts, necessitating further research to improve their reliability. Sung and Shin (2023) evaluated the performance of four publicly available Korean morphological analyzers (*Stanza, Trankit, Kkma, and Komoran*) on L2-Korean written texts. Results showed a reduced performance on the L2 data compared to the L1 data across all investigated analyzers. Specifically, *Stanza*, which demonstrated the highest performance on the L1/L2 datasets (F1: 0.92 [tokenization]; 0.93 [POS tagging]), exhibited reduced performance on the L2 dataset (F1: 0.89 [tokenization]; 0.86 [POS tagging]). Upon a detailed examination of model performance based on by-tag accuracy, it was observed that the POS tags associated with predicates—specifically, VV (Verb), VA (Adjective), and VX (Auxiliary Verb)—showed lower accuracy compared to other tags. Overall, the finding highlights potential challenges involving the use of deep-learning models, trained exclusively on L1 data, to process L2-Korean corpora.

## 2.4 L2 domain-specific model development

Although previous studies have consistently shown that fine-tuning L1 pre-trained models with L2 data improves the accuracy of both tokenization and POS tagging for L2 data (Berzak et al., 2016; Kyle et al., 2022; Sung and Shin, 2023), there are two key questions unresolved with respect to developing L2 domain-specific models. First, it is unclear how the models perform in zero-shot scenarios with unseen L2 data (i.e., L2 test sets not sourced from the same origin as L2 training data[1]), which is a crucial factor for enhancing the model's reliability and robustness (Choi and Palmer, 2012).

Second, the most effective strategy for L2-dataset development, which would yield an optimal L2 model after training, has not been fully identified. This process, particularly the creation of datasets with gold annotations, often requires considerable resources and time (Snow et al., 2008). Specifically in L2 studies, to the best of our knowledge, there is no research determining whether the primary focus in developing an L2 dataset for fine-tuning an L1 pre-trained model should be on increasing the size of the L2 training data or on improving its quality. These two questions invite the need for further investigation, which motivates the current study.

## 3 Experimental Setup

### 3.1 Dataset

We used two L2-Korean and one L1-Korean corpora which include morpheme annotations based on the Sejong tag set (Kim et al., 2007; Appendix A).

**L2-NIKL:** The National Institute of Korean Language (NIKL) compiled a comprehensive L2-Korean learner corpus[2], which consists of written (27,299) and spoken (2,541) texts from years 2015 to 2020. They documented and provided details on learner proficiency in Korean and each participant's L1. The corpus is accessible in raw .txt files and .XML files (with individual morpheme and/or error annotations; see Appendix B as an example), though some data is exclusively in raw format. Data access requires permission[3].

**L2-KLM:** The L2 Korean Learner Morpheme (KLM) corpus consists of 600 L2-Korean written texts, each annotated at the morpheme level. These texts are distributed across six proficiency levels of Korean, with 100 texts per level (Sung and Shin, 2023). The morpheme annotations were manually constructed and underwent comprehensive cross-validation by trained human annotators. The data in this corpus is in CoNLL-U format, adhering to the Universal Dependencies (UD) formalism (Nivre et al., 2020).

---

[1]In this context, "the same origin" refers to learner data from similar settings or conditions, including factors such as the data-collection prompt and/or the nature of the task assigned to L2 Korean learners for language production. This consideration was informed by previous learner corpus studies that show that the task's prompt and instructions can markedly influence learner output (Alexopoulou et al., 2017).

[2]https://kcorpus.korean.go.kr

[3]For more information on this dataset's creation, refer to the official manual from the institute: https://korean.go.kr/front/reportData/reportDataView.do?mn_id=207&searchOrder=years&report_seq=1082&pageIndex=1

**L1-GSD**: The Google Korean Universal Dependency Treebank (UD Korean GSD)[4] was utilized as an L1 reference corpus (Chun et al., 2018; McDonald et al., 2013). This dataset contains roughly 6,000 sentences which were collected from online blogs and news outlets written by Korean native speakers. For evaluation, we used 989 sentences from the UD Korean GSD test set.

## 3.2 Model training and evaluation

We employed Stanza (Qi et al., 2020) (version 1.4.2, Korean-GSD package) for model training and evaluation, which is pre-trained on the UD Korean GSD treebank. It provides a unified pipeline for Korean-specific tasks such as tokenization (based on an eojeol), lemmatization (based on a morpheme), and POS tagging (including both UPOS [Universal POS] and XPOS [treebank-specific POS]). For Korean data, lemmatization and XPOS tagging corresponds to morpheme parsing and morpheme POS tagging, respectively; therefore, these two tasks were utilized. Stanza also offers a separate pipeline for model training tailored for processing CoNLL-U formatted files[5].

We selected a batch size of 5000 for training the POS tagger and 500 for the lemmatizer. Preliminary experiments revealed that a larger batch size of 5000 was most effective for training the POS tagger, whereas the performance for morpheme tokenization remained consistent across different batch sizes (cf. Zhou et al., 2015). In order to optimize the training process, we further used Adam as an optimization algorithm.

To construct and evaluate the model, we randomly split each L2 dataset into three subsets (80% for training; 10% for validation; 10% for test).

## 3.3 Evaluation metrics

For robust evaluation, we measured the model's performance using the following metrics:

- **Perfect score (PS)**: measures the accuracy in morpheme parsing by comparing the number of morphemes within an eojeol. It evaluates the model's ability to predict the exact number of morphemes in alignment with the gold standard.

- **F1**: measures instances of exact matches between predicted tokens/tags and the gold standard, but only when the parsed morpheme count matches the gold-annotated count within an eojeol. This metric aggregates true positives, false positives, and other counts across the dataset, leading to a micro-F1 calculation.

- **By-tag accuracy**: measures correct predictions for each POS tag using micro-F1 scores.

# 4 Experiment 1

The goal of Experiment 1 was to leverage the sizable L2-NIKL dataset to the fullest extent, despite some data loss during the data pre-processing stage.

## 4.1 Data pre-processing

There were several issues in handling the L2-NIKL corpus in its original form. These included format inconsistencies across some files (e.g., files that included only the title of the text without any content), the sentences written by non-Korean characters, and misalignment of sentence-ending punctuation marks and their corresponding POS tags[6].

For the first issue, we extracted files that contained only a single sentence, typically a title, and excluded those. For the second issue, we identified sentences with non-Korean characters, such as Chinese or English alphabets, using the Unicode specification script and excluded them. For the last issue, we treated sentence-final punctuation marks as distinct tokens, placing them at the end of each sentence with their respective POS tags. Furthermore, we replaced all instances of the SYMBOL tag with the SF tag.

After resolving these issues, we extracted individual tokens and their corresponding POS tags, and restructured the entire file format to comply with the CoNLL-U format, while preserving the original metadata[7]. For convenience, we dubbed the L2-NIKL dataset used in Experiment 1 as L2-NIKL-O(riginal) to differentiate it from the dataset used in Experiment 2.

---

[4]https://universaldependencies.org/treebanks/ko_gsd/index.html

[5]https://github.com/stanfordnlp/stanza-train

[6]Based on the UD convention, sentence-final punctuation marks should ideally have been tokenized separately. However, they were instead attached to the sentence-final functional morphemes (e.g., ㄴ다+"."), with the POS tags that are not found in the Sejong tag set (e.g., a period (".") was often incorrectly tagged as SYMBOL instead of the correct SF tag.)

[7]Refer to https://github.com/NLPxL2Korean/KLM-corpus/tree/main/NIKL-corpus-process for the related codes.

Meanwhile, we recognized the need to address the three tags (NF, NV, NA) present in the L2-KLM corpus. Upon reviewing the tags used in the L2-NIKL and L1-GSD datasets, we noticed that these two datasets rarely used the NF (Undecided noun) or NV (Undecided verb) tags. Instead, they they predominantly employed the NA (Undecided) tag to annotate unknown words, most of which originated from spelling errors. We thus merged the NF and NV tags into the NA tag for the L2-KLM dataset for the following training and cross-dataset evaluation.

## 4.2 Data exploration

Table 1 shows the basic statistics of sentences and morphemes from each dataset used in Experiment 1[8]. Table 2 presents the top 15 POS tags frequently occurring in each dataset[9].

| | L2-NIKL-O | L2-KLM | L1-GSD |
|---|---|---|---|
| # sents$_{total}$ | 28,849 | 7,527 | 4,400 |
| # sents$_{written}$ | 15,773 | 7,527 | 4,400 |
| # sents$_{spoken}$ | 13,076 | 0 | 0 |
| # morphemes | 304,501 | 129,784 | 96,028 |

Table 1: Descriptive statistics of the datasets used in Experiment 1

## 4.3 Evaluation and results

Table 3 presents overall PS[10] and F1 of the morphological analyzers for the L2-NIKL-O and L2-KLM test sets. Figure 1 displays the by-tag accuracy for the top 15 POS tags as analyzed in the two L2 test sets.

We note three major findings. First, when the L2-trained models were tested on the data from

| Tag | L2-NIKL-O | L2-KLM | L1-GSD |
|---|---|---|---|
| NNG | 68,223 (22.40) | 25,279 (19.48) | 28,360 (29.53) |
| VV | 26,882 (8.83) | 10,196 (7.86) | 6,561 (6.83) |
| EC | 22,370 (7.35) | 8,558 (6.59) | 7,885 (8.21) |
| EF | 17,604 (5.78) | 7,537 (5.81) | 3,030 (3.16) |
| JKB | 15,185 (4.99) | 6,362 (4.90) | 4,956 (5.16) |
| ETM | 14,862 (4.88) | 6,676 (5.14) | 4,923 (5.13) |
| MAG | 13,807 (4.53) | 4,303 (3.32) | 2,427 (2.53) |
| JX | 13,338 (4.38) | 5,399 (4.16) | 3,946 (4.11) |
| JKO | 10,316 (3.39) | 4,716 (3.63) | 3,185 (3.32) |
| JKS | 10,270 (3.37) | 4,128 (3.18) | 2,527 (2.63) |
| NNB | 9,723 (3.19) | 4,739 (3.65) | 4,004 (4.17) |
| VA | 9,476 (3.11) | 3,375 (2.60) | 1,790 (1.86) |
| XSV | 7,965 (2.62) | 3,267 (2.52) | 3,489 (3.63) |
| NP | 7,436 (2.44) | 2,135 (1.65) | 589 (0.61) |
| VX | 5,322 (1.75) | 3,222 (2.48) | 1,918 (2.00) |
| Total | 252,779 (83.01) | 99,892 (76.98) | 79,590 (82.88) |

Table 2: Frequencies (proportions %) of the top 15 POS tags in the datasets used in Experiment 1

| Training | Metric | L2-NIKL-O | | L2-KLM | |
|---|---|---|---|---|---|
| | | TOK | POS | TOK | POS |
| L2-NIKL-O | PS | **94.32** | 95.10 | 89.64 | 92.01 |
| | PS Δbest | - | ↓1.07 | ↓5.46 | ↓3.18 |
| | F1 | **95.65** | **93.07** | 88.61 | 86.52 |
| | F1 Δbest | - | - | ↓6.70 | ↓5.51 |
| L2-KLM | PS | 92.26 | 93.40 | **95.10** | **95.19** |
| | PS Δbest | ↓2.06 | ↓2.77 | - | - |
| | F1 | 93.60 | 88.88 | **95.31** | **92.03** |
| | F1 Δbest | ↓2.05 | ↓4.19 | - | - |
| L1-GSD | PS | 93.61 | **96.17** | 92.28 | 92.47 |
| | PS Δbest | ↓0.71 | - | ↓2.82 | ↓2.72 |
| | F1 | 95.19 | 88.02 | 92.26 | 87.44 |
| | F1 Δbest | ↓0.46 | ↓5.05 | ↓3.05 | ↓4.59 |

Table 3: Comparison of overall PS and F1 (in %) out of 100 for morphological analyzers on the L2-NIKL-O and L2-KLM test sets. TOK refers to morpheme tokenization (lemmatization in Stanza), while POS refers to morpheme POS tagging. Bold numbers indicate the best score. The symbol Δbest illustrates the difference in performance between the best score and the respective score. A downwards arrow shows decrease in performance compared to the best score.

the same domain as their training, they overall outperformed the L1-GSD baseline model in both L2-NIKL-O and L2-KLM test sets. This is in line with previous findings suggesting that domain-specific data can enhance model performance because of the models' familiarity with patterns found in the target data (e.g., Toutanova et al., 2003; Giménez and Marquez, 2004; Shen et al., 2007). Specifically, the L2 models seem adept at identifying basic words uncommon in the L1 dataset. This is

---

[8] While we did not train a model using L1-GSD, we included information about the L1-GSD training set for reference. This was because the pipeline that we utilized in this study was pre-trained with this dataset.

[9] Tags starting with S, such as SF (sentence-final punctuation marks, e.g., periods), SN (numbers), and SL (foreign characters), were excluded as they were not deemed important for this study.

[10] Ideally, each morpheme should be assigned to its corresponding POS tag, so the PS for TOK and POS should be identical in every instance. In reality, however, the language model in Stanza (and probably other models in other pipelines) has such inconsistencies inherently. To illustrate, the word 불만족한 *pwul-man-cok-han* "unsatisfied" is parsed into four morphemes 불+만족+하+ㄴ but tagged with only three POS tags "NNG+XSA+ETM". While this issue is not the focal investigation point in the present study, future research should seek to find why the original language models manifest the issue and how it can be handled properly.

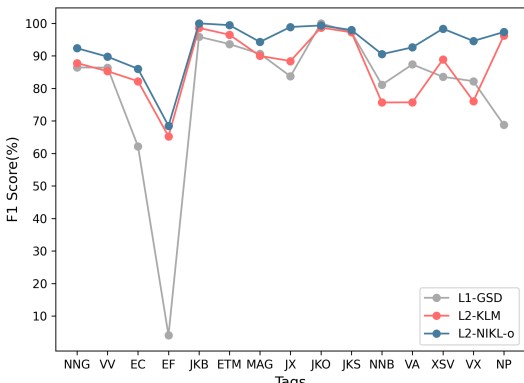

(a) Test set: L2-NIKL-O

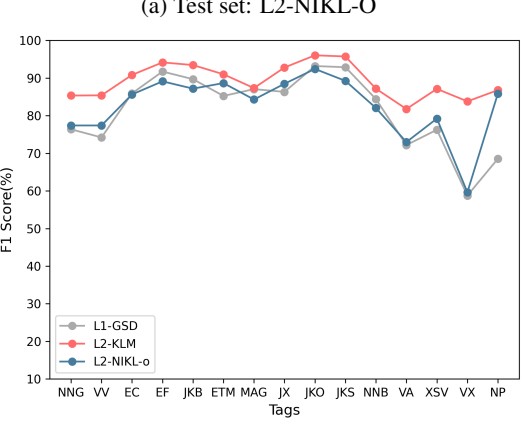

(b) Test set: L2-KLM

Figure 1: By-tag performance of top 15 POS tags in L2-NIKL-O and L2-KLM test sets

supported by the frequent tagging of the noun (NNG) (see Table 2) and the modest increase in F1 for this tag with L2 models (see Figure 1). Furthermore, L2 models may better recognize morpheme combinations prevalent in L2 datasets. This possibly underscores both the Korean language's agglutinative nature in which a word can have multiple morphemes and the fact that (L2-Korean) learners often employ simpler lexico-grammatical structures (Lim et al., 2022; Hwang, 2023) in writing.

Second, an asymmetry was observed in the model's zero-shot performance on unseen L2 data. Both the L2-NIKL-O and L2-KLM models displayed a decline in PS and F1 on the unseen L2 test sets compared to the attested L2 test sets (from the same domain as their training). However, the gap in score reduction[11] was larger for the L2-NIKL-O model than the L2-KLM model.

Third, model performance in the POS-tagging

———————
[11]In most instances, △best represents the difference between a score obtained from a domain-specific test set (which is from the same domain as their training) and a score from zero-shot performance.

task seemed to be heavily influenced by the presence/absence of sentence-final punctuation marks in the L2-NIKL-O dataset. Specifically, the L1-GSD model performed poorly with the sentence-final morpheme (EF) when handling the L2-NIKL-O test set (as in Figure 1a). This observation prompted a more detailed examination of the L2-NIKL-O dataset and its POS-tagging outcomes. We observed that the EF tag was frequently mislabeled as EC (Ending, connecting), particularly when the sentence-final punctuation mark was absent in the L2-NIKL-O test set. In the L2-NIKL-O training set, a similar issue arose with certain sentences, notably those from spoken sources, lacking sentence-final punctuation marks.

The last two findings suggest that thorough data cleaning, as seen in the creation of the hand-crafted L2-KLM corpus, is crucial when fine-tuning an L2 model. In other words, the most promising L2 model would be something that does not forget what it has learned from the L1 training data (Kirkpatrick et al., 2017) and is exposed to robust representations of L2 data. This led us to the next stage of this study, which involved further cleaning of the L2-NIKL-O dataset and additional training/evaluation of the L2 model to assess its robustness.

## 5 Experiment 2

The goal of Experiment 2 was to address potential issues inherent in the L2-NIKL-O dataset, which may have aggravated the performance of the model. Therefore, we undertook further data-cleaning as outlined below.

### 5.1 Data cleaning

First, we excluded sentences without sentence-final punctuation marks. We then eliminated one-token responses from the spoken data due to their repetitive nature and potential to introduce noise during model tuning. Examples include the words like 네 *ney* "yes" or 아 *ah* "ah (interjection)". Lastly, we omitted sentences with morphemes tokenized from eojeols but missing their corresponding POS tags. Given the likely unreliability and incorrect tagging of these sentences, their inclusion could diminish fine-tuning quality.

For convenience, we dubbed the L2-NIKL dataset in Experiment 2 as L2-NIKL-C(leaned) to differentiate it from the L2-NIKL-O dataset used in Experiment 1.

## 5.2 Data exploration

Figure 2 visualizes the length of sentences in each dataset, determined by the count of eojeols per sentence. The L2-NIKL-C dataset included a smaller number of short sentences than the L2-NIKL-O dataset as a result of the additional data-cleaning process.

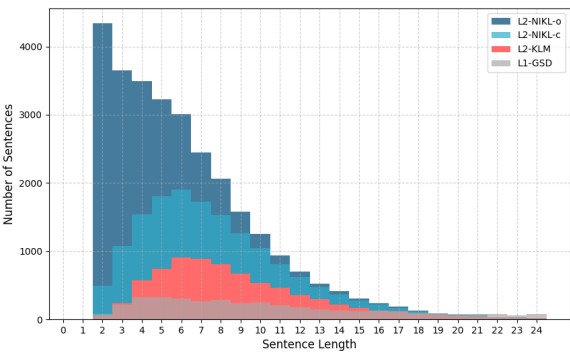

Figure 2: Visualization of sentence lengths in each dataset

In parallel to Table 1 and 2, Table 4 shows the basic statistics of sentences and morphemes from the Experiment 2 datasets. Table 5 illustrates the 15 most frequent POS tags in each dataset. Overall, the size of the L2-NIKL-C dataset was approximately half of the L2-NIKL-O dataset while keeping the POS-tag distributions compatible with each other.

| | L2-NIKL-C | L2-NIKL-O |
|---|---|---|
| # sents$_{total}$ | 14,682 | 28,849 |
| # sents$_{written}$ | 12,973 | 15,773 |
| # sents$_{spoken}$ | 1,709 | 13,076 |
| # morphemes | 201,920 | 304,501 |

Table 4: Descriptive statistics of the datasets used in Experiment 2

## 5.3 Evaluation and results

Table 6 presents the comprehensive PS and F1 of the morphological analyzers evaluated on the L2-NIKL-C and L2-KLM test sets. Figure 3 shows by-tag accuracy for the top 15 POS tags as analyzed in the L2-NIKL-C test set.

Consistent with the findings from Experiment 1, models tuned on L2 data outperformed the others when tested within the same training domain. Additionally, both the L2-NIKL-C and L2-KLM models saw a decrease in PS and F1 in zero-shot

| Tag | L2-NIKL-C | L2-NIKL-O |
|---|---|---|
| NNG | 45,444 (22.51) | 68,223 (22.40) |
| VV | 18,562 (9.19) | 26,882 (8.83) |
| EC | 14,576 (7.22) | 22,370 (7.35) |
| EF | 14,231 (7.05) | 17,604 (5.78) |
| JKB | 10,621 (5.26) | 15,185 (4.99) |
| ETM | 10,131 (5.02) | 14,862 (4.88) |
| MAG | 8,693 (4.31) | 13,807 (4.53) |
| JX | 8,909 (4.41) | 13,338 (4.38) |
| JKO | 7,796 (3.86) | 10,316 (3.39) |
| JKS | 7,358 (3.64) | 10,270 (3.37) |
| NNB | 6,667 (3.30) | 9,723 (3.19) |
| VA | 6,643 (3.29) | 9,476 (3.11) |
| XSV | 5,843 (2.89) | 7,965 (2.62) |
| NP | 3,858 (1.91) | 7,436 (2.44) |
| VX | 4,355 (2.16) | 5,322 (1.75) |
| Total | 173,687 (86.08) | 252,779 (83.01) |

Table 5: Frequencies (proportions %) of the top 15 POS tags in the datasets used in Experiment 2

tests. Yet, the gap in score reduction was largely similar between the two models. Notably, it was somewhat surprising that the model fine-tuned on L2-KLM surpassed the L1-GSD baseline model in POS tagging for the unseen L2-NIKL-C test set. This indicates that models tuned on L2 data might be more effective in handling L2 test data, even in a zero-shot scenario, compared to models exclusively trained on L1 data.

| Training | Metric | L2-NIKL-C | | L2-KLM | |
|---|---|---|---|---|---|
| | | TOK | POS | TOK | POS |
| L2-NIKL-C | PS | **97.67** | **98.00** | 91.64 | 92.07 |
| | PS Δbest | - | - | ↓3.46 | ↓3.12 |
| | F1 | **98.63** | **96.91** | 91.13 | 86.45 |
| | F1 Δbest | - | - | ↓4.18 | ↓5.58 |
| L2-KLM | PS | 94.01 | 94.45 | **95.10** | **95.19** |
| | PS Δbest | ↓3.66 | ↓3.55 | - | - |
| | F1 | 95.26 | 92.14 | **95.31** | **92.03** |
| | F1 Δbest | ↓3.37 | ↓4.77 | - | - |
| L1-GSD | PS | 94.45 | 94.34 | 92.28 | 92.47 |
| | PS Δbest | ↓3.22 | ↓3.66 | ↓2.82 | ↓2.72 |
| | F1 | 95.15 | 88.67 | 92.26 | 87.44 |
| | F1 Δbest | ↓3.48 | ↓8.24 | ↓3.05 | ↓4.59 |

Table 6: Comparison of overall PS and F1 (in %) out of 100 for morphological analyzers on the L2-NIKL-C and L2-KLM test sets

A detailed examination of Figure 3 reveals that the by-tag accuracy exhibited larger stability in the L2-KLM model than the L1-GSD baseline model. This is particularly noticeable for ETM (Ending, de-

terminative), JX (Postposition, auxiliary), and NP (Pronoun) tags. Conversely, the L2-KLM model demonstrates somewhat reduced performance on VA (Adjective) and VX (Verb, auxiliary) tags. This is attributable to the robust and fine-grained annotation scheme employed in the KLM annotation process (cf. Sung and Shin, 2023, p.3), which treats predicate-related morphemes as an important piece of morpho-syntactic knowledge in acquiring Korean.

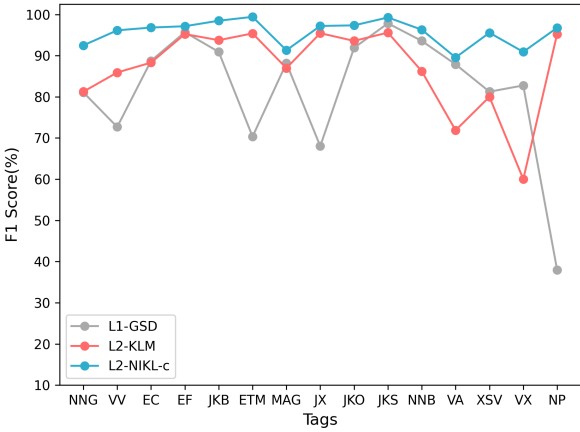

Figure 3: By-tag performance of top 15 POS tags in the L2-NIKL-C test set

Taken together, the findings of Experiments 1 and 2 suggest that merely increasing the size of training data does not necessarily enhance performance for fine-tuned L2 pipelines. Instead, the quality of the data, which encompasses its annotation and cleaning, plays an important role. Specifically, even though the L2-NIKL-O training set (304,501 tokens) was larger than that of the L2-KLM model (129,784 tokens), the L2-NIKL-O model did not outperform the L2-KLM model in zero-shot tests, including those on the L1 test set (Table 7). In a similar vein, the L2-NIKL-C model, with its training set of approximately half the size (201,920 tokens) of the L2-NIKL-O model, did not lag behind in performance. The performance rather improved after we cleaned the NIKL dataset for training, especially in terms of tokenization.

## 6    Conclusion and Future Works

Through the two experiments above, we have revealed three key aspects with regard to constructing L2 models. First, models fine-tuned on L2 data can outperform an L1 baseline when evaluated within the same domain. Second, model performance can

| Training | Metric | **L1-GSD TOK** | **L1-GSD POS** |
|---|---|---|---|
| L1-GSD | PS | **94.40** | **95.04** |
| | PS Δbest | - | - |
| | F1 | **94.51** | **93.65** |
| | F1 Δbest | - | - |
| L2-KLM | PS | 86.97 | 88.92 |
| | PS Δbest | ↓7.43 | ↓6.12 |
| | F1 | 88.58 | 83.40 |
| | F1 Δbest | ↓5.93 | ↓10.25 |
| L2-NIKL-C | PS | 88.05 | 89.51 |
| | PS Δbest | ↓6.35 | ↓5.53 |
| | F1 | 87.71 | 81.15 |
| | F1 Δbest | ↓6.80 | ↓12.50 |
| L2-NIKL-O | PS | 83.24 | 88.96 |
| | PS Δbest | ↓11.16 | ↓6.08 |
| | F1 | 81.41 | 80.96 |
| | F1 Δbest | ↓13.10 | ↓13.00 |

Table 7: Comparison of overall PS and F1 (in %) out of 100 for morphological analyzers on the L1-GSD (reference) test set

exhibit asymmetry and variability in zero-shot evaluations. Lastly, augmenting the size of the training dataset does not always lead to enhancing the performance of a fine-tuned model. The findings of this study thus confirm domain-specificity in light of data processing and highlight the importance of input quality, including rigorous annotations (conducted by human annotators as in the case of the L2-KLM) or cleaning (as exemplified in the case of the L2-NIKL-C) when fine-tuning L2 models on the basis of existing L1-oriented, pre-trained models.

Effectively augmenting computational resources for lesser-studied languages and language-usage contexts requires meticulously engineered and validated data-processing pipelines. This is done not only by expanding target L2 corpora with gold annotations but also by refining existing tagging schemes. For example, in our study, the L2-NIKL corpus, while robust in quantity, lacks clear guidelines on whether annotations were manually created from scratch or generated with the assistance of automatic taggers[12]. Building upon the find-

---

[12] According to the official NIKL guideline[2], Korean-specific morpheme analyers (i.e., *Kkma, Komoran, ETRI, Khaiii, KIWI, Mecab-Korean, Utagger*) were evaluated in terms of their performance in order to facilitate the creation of sizable L2-Korean morpheme annotations (p. 188). However, it remains unclear if NIKL utilized semi-automatic annotations, which involve automatic analysis followed by human verification, or if they adopted different approaches to this task. This ambiguity becomes problematic, especially when addressing tagging errors made by L2 learners. L1-pre-trained models often face difficulties in identifying these errors, mostly by incorrectly assigning them one of the most frequent POS tags

ings of this study, researchers are encouraged to consider improving the quality of gold annotations for more reliable and robust results in automatic handling of L2 data.

Beyond the primary focus of this study on morpheme tokenization and POS tagging, extending the scope of investigation toward such tasks as dependency parsing would offer a holistic evaluation of automatic analyses on learner corpora. We believe this extension will contribute to not only promoting a sense of DEI in research but also facilitating AI literacy—by improving researchers' understanding of how computational algorithms operate and how these can be applied to research purposes appropriately. Moreover, pursuing this research direction would advance NLP applications in educational contexts, offering valuable resources on language learning and teaching.

## Limitations

Although we pre-processed the L2-NIKL data following the outlined steps, we acknowledge that there may be different, and potentially more optimal, methods for data pre-processing. Additional experiments would yield a more comprehensive report on model performance, along with strengths and weaknesses of the model. The L2-NIKL data is substantially restricted in its use because of the major drawbacks inherent in the data themselves (e.g., unclear data-collection process, little control for topics and prompts, imbalanced distribution of learners' L1 backgrounds). Furthermore, since the L2-NIKL data cannot be shared without permission, this limits the replicability of this study, deviating from Open Science practices.

## Ethics Statement

We used two publicly available datasets (L1-GSD and L2-KLM) and one dataset with permission required (L2-NIKL). For the two open-access datasets, we ensured that all potentially identifying information such as names and other personal details was removed. For the permission-only dataset, we strictly followed all stipulated guidelines to respect the interests of the data providers. Other than that, we believe that there is no substantial ethical issue with the research presented in this study. There is no dishonesty in the execution

or presentation of the research, including plagiarism, deliberate violation of anonymity, citation, review, or duplicate submission policies, falsifying results, or misrepresentation. Our research is expected to broaden the horizon of language science at the interface of technology, currently being limited to a handful of the world's over 7000 languages (Joshi et al., 2020), which contributes to promoting more inclusive research practice toward diverse languages and language-usage contexts.

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

## A  Sejong tag set

Table 8 provides a description of the Sejong tag set used in this study (Kim et al., 2007).

| Tag | Description |
|-----|-------------|
| NNG | Noun, common (보통 명사) |
| NNP | Noun, proper (고유 명사) |
| NNB | Noun, common bound (의존 명사) |
| NR | Numeral (수사) |
| NP | Pronoun (대명사) |
| VV | Verb, main (동사) |
| VA | Adjective (형용사) |
| VX | Verb, auxiliary (보조 동사) |
| VCP | Copular, positive (긍정 지정사) |
| VCN | Copular, negative (부정 지정사) |
| MM | Determiner (관형사) |
| MAG | Common adverb (일반 부사) |
| MAJ | Conjunctive adverb (접속 부사) |
| IC | Exclamation (감탄사) |
| JKS | Postposition, nominative (주격 조사) |
| JKC | Postposition, complement (보격 조사) |
| JKG | Postposition, prenominal (관형격 조사) |
| JKO | Postposition, objectival (목적격 조사) |
| JKB | Postposition, adverbial (부사격 조사) |
| JKV | Postposition, vocative (호격 조사) |
| JKQ | Postposition, quotative (인용격 조사) |
| JC | Postposition, conjunctive (접속 조사) |
| JX | Postposition, auxiliary (보조사) |
| EP | Ending, prefinal (선어말 어미) |
| EF | Ending, closing (종결 어미) |
| EC | Ending, connecting (연결 어미) |
| ETN | Ending, nounal (명사형 전성 어미) |
| ETM | Ending, determinitive (관형형 전성 어미) |
| XPN | Prefix, nounal (체언 접두사) |
| XSN | Suffix, noun derivative (명사 파생 접미사) |
| XSV | Suffix, verb derivative (동사 파생 -) |
| XSA | Suffix, adjective derivative (형용사 파생 -) |
| XR | Root (어근) |
| NA | Undecided (분석 불능) |
| SF | Period, Question, Exclamation (마침표 등) |
| SE | Ellipsis (줄임표) |
| SS | Quotation, Bracket, Dash (따옴표 등) |
| SP | Comma, Colon, Slash (쉼표, 콜론, 빗금) |
| SO | Hyphen, Swung Dash (붙임표, 물결표) |
| SW | Symbol (기타기호) |
| SH | Chinese characters (한자) |
| SL | Foreign characters (외국어) |
| SN | Number (숫자) |

Table 8: Sejong tag set

## B  L2-NIKL morpheme annotation

Figure 4 provides an example of the L2-NIKL morpheme annotation in an XML file.

```
<Korean_Learners_Corpus>
  <Header>
    <SampleSeq>131</SampleSeq>
    <LearningEnvironment>국내</LearningEnvironment>
    <SourceType>문어</SourceType>
    <AssignmentType>시험 작문</AssignmentType>
    <AssignmentGenre>생활문</AssignmentGenre>
    <AssignmentTheme>살고 싶은 집</AssignmentTheme>
    <SentenceCount>11</SentenceCount>
    <WordCount>72</WordCount>
    <LearnerInfo sequence="1">
      <LearnerType>일반</LearnerType>
      <LearnerTypeDetail>일반</LearnerTypeDetail>
      <AgeGroup>20</AgeGroup>
      <DataGrade>1급</DataGrade>
      <Nationality>중국</Nationality>
      <LearningPurpose>진학</LearningPurpose>
      <MotherLanguage>중국어</MotherLanguage>
    </LearnerInfo>
  </Header>
  <BODY>
    <P to="9" from="0">
      <SENTENCE to="9" from="0">
        살고 싶은 집
        <MorphemeAnnotations>
          <word>
            <w>살고</w>
            <morph from="0" to="2" analyzedType="Ambiguity" pos="VV" subsequence="1">살</morph>
            <morph from="0" to="2" analyzedType="Ambiguity" pos="EC" subsequence="2">고</morph>
          </word>
          <word>
            <w>싶은</w>
            <morph from="3" to="5" analyzedType="Ambiguity" pos="VX" subsequence="1">싶</morph>
            <morph from="3" to="5" analyzedType="Ambiguity" pos="ETM" subsequence="2">은</morph>
          </word>
          <word>
            <w>집</w>
            <morph from="6" to="7" analyzedType="Ambiguity" pos="NNG" subsequence="1">집</morph>
          </word>
        </MorphemeAnnotations>
      </SENTENCE>
    </P>
    <P to="49" from="9">
      <SENTENCE to="49" from="9">
        저는 방 3개, 큰 거실 하나, 화장실 2개가 있는 집에 살고 싶다.
        <MorphemeAnnotations>
          <word>
            <w>저는</w>
            <morph from="9" to="11" analyzedType="Ambiguity" pos="NP" subsequence="1">저</morph>
            <morph from="9" to="11" analyzedType="Ambiguity" pos="JX" subsequence="2">는</morph>
          </word>
          <word>
            <w>방</w>
            <morph from="12" to="13" analyzedType="Ambiguity" pos="NNG" subsequence="1">방</morph>
          </word>
          <word>
            <w>3개,</w>
            <morph from="14" to="17" analyzedType="Normal" pos="SYMBOL" subsequence="1">3</morph>
            <morph from="14" to="17" analyzedType="Ambiguity" pos="NNB" subsequence="2">개</morph>
            <morph from="14" to="17" analyzedType="Normal" pos="SP" subsequence="3">,</morph>
          </word>
          <word>
```

Figure 4: An example of the L2-NIKL morpheme annotation

## C Hyperparameter values

Table 9 and Table 10 provide information about the hyperparameter settings for lemmatization (morpheme tokenization) and POS tagging trainings.

| Hyperparameter | Selected |
|---|---|
| Num hidden units | 200 |
| Embedded vector space | 50 |
| Number of layers | 1 |
| Dropout rate of layers | 0.5 |
| Beam size | 1 |
| Attension type | soft |
| Optimization algorithm | Adam |
| Learning rate | 0.001 |
| Numepoch | 60 |
| Batch size | 50 |
| Max grad norm | 5.0 |

Table 9: Hyperparameters used for lemmatization

| Hyperparameter | Selected |
|---|---|
| Batch size | 5000 |
| Max training steps | 50000 {32, 64, 128, 50000} |
| Learning rate | 3e-3 |
| Optimization algorithm | Adam |
| Max grad norm | 5.0 |

Table 10: Hyperparameters used for POS tagging