# OpenReview forum: "Diversifying language models for lesser-studied languages and language-usage contexts: A case of second language Korean"
_EMNLP/2023/Conference — EMNLP 2023 Findings_

### Official Review · Reviewer_i3cX · 2023-08-04

**Soundness:** 4

**Excitement:**

3: Ambivalent: It has merits (e.g., it reports state-of-the-art results, the idea is nice), but there are key weaknesses (e.g., it describes incremental work), and it can significantly benefit from another round of revision. However, I won't object to accepting it if my co-reviewers champion it.

**Paper Topic And Main Contributions:**

The authors present a study about morpheme POS tagging and parsing of L2 Korean speakers. They conclude that models trained on L2 data can outperform models trained on L1 data when tested on the same domain. Second, more data is not always better, as cleaned training sets of L2 speakers provide a better result than the original training set with more samples. Third, performance is not bidirectional when testing on unseen L2 data from different domains.

Overall, this paper is complete and proposes an interesting and solid study about Korean language learners text.

**Questions For The Authors:**

- Have you considered the effect of the native language of the L2 speakers? This probably has an effect on the characteristics of L2 sentences.
- I do not really understand the benefits of L2 models when tested out-domain, as they perform evenly or worse than the baseline. What are the advantages in this case, if any?

**Reasons To Accept:**

- Code will be public.
- Complete and exhaustive experimental setup: extensive experiments and detailed discussion.
- In-depth data treatment, both analysis and cleaning.

**Reasons To Reject:**

- None

**Reproducibility:**

5: Could easily reproduce the results.

**Reviewer Confidence:**

2: Willing to defend my evaluation, but it is fairly likely that I missed some details, didn't understand some central points, or can't be sure about the novelty of the work.

**Typos Grammar Style And Presentation Improvements:**

The titles in English are usually written capitalising all words except for short articles, prepositions, and coordinating conjunctions. For example, the title of the paper should be: "Diversifying Language Models for Lesser-Studied Languages and Language-Usage Contexts: A Case of Second Language Korean.”

---

> ### Author Rebuttal · Authors · 2023-08-28
>
> **Reviewer 3**
>
> We sincerely appreciate the Reviewer 3’s insightful questions and suggestion regarding the title style. We have addressed each comment in the sections below:
>
> ---
>
> **1. (Question):** Have you considered the effect of the native language of the L2 speakers? This probably has an effect on the characteristics of L2 sentences.
>
>   **Response:**
>   **The influence of the native language on L2 writing is undeniably significant in L2 research, but it was not the central point of our current study.** This was because we were primarily interested in how currently available language models work with L2 data processing in a broad sense. Together with providing detailed background information (e.g., proficiency level, L1 background, etc.) on L2 speakers from both datasets (i.e., L2-NIKL; L2-KLM) in the appendices, we have mentioned that future research (with a more sizeable dataset) should seek to pursue this issue thoroughly.
>
> ---
>
> **2. (Question):** I do not really understand the benefits of L2 models when tested out-domain, as they perform evenly or worse than the baseline. What are the advantages in this case, if any?
>
>   **Response:** While it might appear at first glance that L2 models perform similarly or even worse than the baseline in some instances (as in Table 3), **it is important to highlight that we observed improvements in certain POS tags, as depicted in Figures 1 and 3. Moreover, our data from Lines 466-473 revealed an intriguing trend such that the models trained on the L2-KLM dataset surpassed the performance of the L1-GSD baseline model in POS tagging of the out-domain (unseen) L2-NIKL-c test set (also illustrated in Table 6)**. This suggests that a model fine-tuned with L2 data might be better equipped to handle out-domain L2 data than that solely reliant on L1 data (the baseline model). Thus, our findings underscore the potential necessity of fine-tuning current L1 models to L2-specific domains. Further emphasizing this point, our discussions in Section 6 (Lines 514-521) advocate for a heightened focus on input quality, emphasizing the imperative for rigorous annotations and thorough data cleaning. We believe that such measures are integral for developing L2 models that benefit overarching L2 studies, not just those confined to L2 Korean.
>
> ---
>
> **3. (Style):** The titles in English are usually written capitalising all words except for short articles, prepositions, and coordinating conjunctions. For example, the title of the paper should be: "Diversifying Language Models for Lesser-Studied Languages and Language-Usage Contexts: A Case of Second Language Korean.”
>
>   **Response:** We have made the necessary changes to ensure that the title adheres to the convention.

---

### Official Review · Reviewer_qhvV · 2023-08-05

**Soundness:** 3

**Excitement:**

3: Ambivalent: It has merits (e.g., it reports state-of-the-art results, the idea is nice), but there are key weaknesses (e.g., it describes incremental work), and it can significantly benefit from another round of revision. However, I won't object to accepting it if my co-reviewers champion it.

**Paper Topic And Main Contributions:**

This paper mainly investigates the generalizability of the morphological parsers/taggers on second-language (L2) Korean datasets. The results underline the importance of the quality of training data; the model trained on cleaned training data outperforms the one trained on original training data even though the size of cleaned training data is half of the original size.

**Reasons To Accept:**

This paper shows an interesting result: increasing the size of training data does not always help to enhance performance. The result can be an essential instruction for developing L2 Korean data.

**Reasons To Reject:**

My main concern about this paper is that the motivation is somewhat unclear to me even after reading Section 2. Concretely, the paper did not explain why it chose L2 Korean to investigate the generalizability of models. It would be better to use some examples or statistics that can show the different natures of L1 and L2 data.

**Reproducibility:**

5: Could easily reproduce the results.

**Reviewer Confidence:**

4: Quite sure. I tried to check the important points carefully. It's unlikely, though conceivable, that I missed something that should affect my ratings.

---

> ### Author Rebuttal · Authors · 2023-08-28
>
> **Reviewer 2**
>
> We sincerely appreciate the Reviewer 2’s feedback and concern.
>
> ---
>
> **1. (Feedback/concern):** My main concern about this paper is that the motivation is somewhat unclear to me even after reading Section 2. Concretely, the paper did not explain why it chose L2 Korean to investigate the generalizability of models. It would be better to use some examples or statistics that can show the different natures of L1 and L2 data.
>
>   **Response:**
>   **Our primary focus centers upon L2 Korean studies. However, we believe this research would appeal to computational linguists who prioritize a sense of Diversity, Equity, and Inclusion in research because L2 Korean represents one of the lesser-studied languages and language-usage contexts in the field.**
>
>   To provide clarity for Section 2, we elaborated on the motivation for our study. We began by explaining the agglutinative nature of Korean, characterized by its morphological intricacies (Section 2.1). These complexities are often amplified when encountered by L2 learners. Meanwhile, a growing number of L2-Korean studies have started utilizing automatic morphological parsers/taggers, predominantly in quantitative research (Section 2.2). However, when existing models – trained exclusively on L1 datasets – are applied to L2 Korean data, they have shown notable discrepancies in performance (Section 2.3). These computational challenges, combined with the growing demand on L2-specific models from (applied) linguistics, underscored the need for models tailored to the L2-Korean domain. Furthermore, we drew attention to the ongoing research efforts in L2 English and Chinese in Section 1 (Lines 47-48: Kyle et el., 2022; Lan et al., 2023 – Upon reviewing this part, we realized it might be beneficial to specify that the studies by Kyle et al. (2022) and Lan et al. (2023) are related to L2 English and L2 Chinese, respectively).

---

### Official Review · Reviewer_prvE · 2023-08-05

**Soundness:** 2

**Excitement:**

3: Ambivalent: It has merits (e.g., it reports state-of-the-art results, the idea is nice), but there are key weaknesses (e.g., it describes incremental work), and it can significantly benefit from another round of revision. However, I won't object to accepting it if my co-reviewers champion it.

**Paper Topic And Main Contributions:**

This paper presents a mostly experimental contribution, focusing on morphological analysis of L2 Korean and its specificities compared to L1 Korean (hence a lesser-studied language + a lesser-studied language-usage context). Two sets of experiments are described:
- one about fine-tuning an L1 model with L2 data and studying the impact of train-test domain discrepancy even within L2 data,
- and another one that is more oriented towards data quality, cleaning annotation artefacts and where the authors analyze the impact of that cleaning on domain mismatch effects.

The authors also intend to share the script for cleaning that data, which by itself is a contribution as it can facilitate future research on L2 Korean and its morphological analysis.

**Questions For The Authors:**

A - Line 215, it would be useful to expand a bit on what is meant by "same origin" (same speakers?) and why it creates this uncertainty (I assume this is because some L2 errors vary a lot from one speaker to another, so the dataset leaks with respect to the typology of errors? Is that all?).

B - Line 301: Unclear how "Perfect score" is defined. "comparing the number" is dubious (does not seem to match what is done in the provided reference): only the number, or their boundaries as well? And how is this defined when using Perfect score on the POS task? This is important information that should be clear in the text itself (not relying solely on the reference).

C - Line 305: Unclear how the F1 score is defined. Instances=the morphemes? What is the difference between "parsing is perfect" and "tokens align exactly with the golds" (the metric seems tautological)? And the phrasing looks more as an accuracy than an F1 (what do precision and recall mean, with that definition?). For the POS task, is it micro-F1 or macro-F1?

D - Line 326: is it "consist" (only non-Korean) or "contain" (some non-Korean)? Line 320 seemed to imply "contain"... For instance, if a Korean sentence happens to contain the name of a foreign entity, written in Latin script, is the sentence discarded? This warrants a bit more discussion, at least for more clarity on what has been done, and preferably to explain why it is the right way to do that (or why it is not ideal and only chosen for practical reasons, but it is still OK and does not hurt representativeness because of this or that reason).

**Reasons To Accept:**

Lesser-studied languages are a known important topic, but this paper further expands it by introducing the consideration of language-usage context. This triggers new ideas, and could lead to expanding the views of many researchers. Also, the paper itself offers a lot of context (§2 provides an interesting review of L2-Korean and the associated challenges and applications), and the experimental work is conducted with rigorous analysis. In particular, it is appreciable to see how the authors performed error analysis and examined detailed results to better understand what phenomena affected the performance of the models (around line 390, line 482...).

Suggestion: Line 581, "expected to ameliorate research biases", I think this could be expanded to put more light on how this contribution is beneficial from that ethical point of view (because it is!).

**Reasons To Reject:**

The main two shortcomings are:
- a) Lack of clarity on the evaluation metrics (see questions below), which renders the numbers quite hard to interpret.
- b) There is a discrepancy between the initial framing of the paper (title, introduction...) which is about lesser-studied languages and language-usage contexts, and the actual content of the paper which is solely about morphological analysis of L2-Korean. Is it a paper about low-resourced languages/domains (and Korean is just a case study to expand upon), or is it a paper about morphological analysis? If the former, it would be really useful for instance to add a new section "Discussion" before the conclusion, in order to discuss how that case study can be generalized to the targeted (broader) problem, which findings are new and reusable for low-resourced languages/domains in general... based on specific facts, and well-articulated.

There are also a few places where the authors make claims or conclusions that appear rather void:
- Line 470-473: yes, matching the domain of training and test data is at the core of machine learning, and using in-domain data to fine-tune a model trained on out-of-domain data is the basis of many domain adaptation approaches, so "implies the possibility that may..." is awkward. What is the real conclusion here, what is the discovery?
- Line 372: this is certainly ill-phrased, because "patterns found in the data" is really what machine learning is supposed to do in general... We do want to recognize patterns, otherwise there is nothing to learn. Question is: are those appropriate patterns, are they informative for that language/domain, is the dataset leaking, etc. This should be rephrased and expanded to better delineate what phenomena are specifically targeted here.

**Reproducibility:**

3: Could reproduce the results with some difficulty. The settings of parameters are underspecified or subjectively determined; the training/evaluation data are not widely available.

**Reviewer Confidence:**

4: Quite sure. I tried to check the important points carefully. It's unlikely, though conceivable, that I missed something that should affect my ratings.

**Typos Grammar Style And Presentation Improvements:**

- At the end of §1 it would be good to have some teaser on the outline, so that the readers can identify more directly where to find the information they seek. This can be short, for instance in the style of "After introducing the specificities of L2-Korean (§2), we present in §3...".
- Title of §3: this section is not really about a method, it discusses data, metrics... basically, the experimental conditions. A more usual title for a section like this would be "Experimental setup".
- Title of §3.2: this section is also about evaluation, so titling it "Model training and evaluation" would be more appropriate.
- Line 298: missing word after "performance"? "model performance measure was undertaken"?
- Titles of §4 and §5: any way to have a more explicit title than "Experiment N"? Something that actually summarizes the purpose or conditions of that experiment. That would be helpful to understand the outline, but also in other places of the paper that make references to those experiments, it would certainly help to understand what is referred to and why.
- Line 366 (and same for lines 458, 471, and probably others): It would be better to use "tuned" rather than "trained" in order to better distinguish models trained from scratch from models fine-tuned on that data. A phrasing like "the models trained on L2 data" seems to refer to models which have never seen any L1 data.
- Footnote 9: issues with parentheses and commas. After "S": missing comma + remove parenthesis. Before "SN": missing comma.
- Table 3 (and same for Table 4): The way to read the table is really not obvious (why that many numbers in the same cell), the answer is only given in the caption and part of it is given only implicitly (by means of putting Perfect in parentheses). I would suggest reducing the first column (maybe rotating the labels? or just reducing fontsize?) so that a short second column can be added to indicate the meaning of each line, i.e. adding for instance "F1" / "\Delta_{best}" / "PS" / "\Delta_{best}" vertically just on the right of "L2-NIKL-o Train".
- Table 3's caption (and same for Table 4's caption): missing final dot
- Line 378 (and same for line 462): "Perfect" with capital P would render the sentence more legible.
- Line 442: "presents shows" --> "shows"
- Better to reference the annexes in the main text (explicit pointers from the sections they provide more details for), so that the reader knows immediately while reading the main text when additional information is available elsewhere.

---

> ### Author Rebuttal · Authors · 2023-08-28
>
> **Reviewer 1**
>
> We sincerely appreciate Reviewer 1 for the many helpful comments and suggestions. We have addressed each comment below and highlighted the changes we plan to make in the revised manuscript. We’ve reordered the suggestions/questions by grouping related comments.
>
> ---
>
> **1. (Suggestion):** Line 581, "expected to ameliorate research biases", I think this could be expanded to put more light on how this contribution is beneficial from that ethical point of view (because it is!).
>
>   **Response:** To better (and more straightforwardly) articulate this point, we have revised the sentence in line 581 as follows:
>
>   “Our research is expected to broaden the horizon of language science at the interface of technology, currently being limited to a handful of the world’s over 7000 languages (Joshi et al., 2020), which contributes to promoting more inclusive research practice toward diverse languages and language-usage contexts”.
>
> ---
>
> **2. (Shortcoming):**  There is a discrepancy between the initial framing of the paper (title, introduction...) which is about lesser-studied languages and language-usage contexts, and the actual content of the paper which is solely about morphological analysis of L2-Korean. Is it a paper about low-resourced languages/domains (and Korean is just a case study to expand upon), or is it a paper about morphological analysis? If the former, it would be really useful for instance to add a new section "Discussion" before the conclusion, in order to discuss how that case study can be generalized to the targeted (broader) problem, which findings are new and reusable for low-resourced languages/domains in general... based on specific facts, and well-articulated.
>
> **Response:**  Our focus was predominantly on L2 Korean as a case study. While the broader context of lesser-studied languages and language-usage contexts served as an important background of this research, the core motivation was to delve into the intricacies of morphological analysis pertaining to L2 Korean. **In other words, our primary research aim was not to generalize our findings to other low-resource languages or domains. Instead, we sought to provide an empirical study on L2 Korean which, we believe, may offer insights for both researchers in computational linguistics aiming for diversification in their model across languages / domains and those in L2 learning-teaching leveraging computational tools.**
>
> **Meanwhile, we may modify the title and introduction in a way that better aligns with the content.** A potential title would be: “Morphological Analysis of L2 Korean: Experimental Insights into Developing Domain-Specific Model”. The revised introduction will emphasize the paper’s focus on morphological analysis of L2 Korean rather than its broader context of low-resource languages and domains. We would also elaborate on potential contributions on lesser-studied languages and language-usage contexts in the Ethics Statement section.
>
> ---
>
> **3. (Shortcoming; Lines 470-473):** yes, matching the domain of training and test data is at the core of machine learning, and using in-domain data to fine-tune a model trained on out-of-domain data is the basis of many domain adaptation approaches, so "implies the possibility that may..." is awkward. What is the real conclusion here, what is the discovery?
>
> **Response:**  We acknowledge that the original phrasing was ambiguous and did not straightforwardly convey our findings. We intended to claim that the experiment demonstrated the effectiveness of using in-domain data to fine-tune a model initially trained on out-of-domain data. **Based on the suggestion, we have revised this part in the following way to provide a more straightforward interpretation of our results.**: “This suggests that a model trained on L2 data is more effective for handling L2 test data than a model solely trained on L1 data.”
>
> ---
>
> **4. (Shortcoming; Line 372):** this is certainly ill-phrased, because "patterns found in the data" is really what machine learning is supposed to do in general... We do want to recognize patterns, otherwise there is nothing to learn. Question is: are those appropriate patterns, are they informative for that language/domain, is the dataset leaking, etc. This should be rephrased and expanded to better delineate what phenomena are specifically targeted here.
>
> **Response:** We admit that our initial wording was overly generic and did not clearly delineate the specific phenomena that we wanted to highlight. **The followings are two major aspects considered in our study.**
>
> **First, the L2-domain specific model learned basic words that are rarely found in the L1 training dataset.** As shown in Table 2, noun (tagged as NNG) emerged as the most frequently identified POS tag in the datasets, and the F1 scores for predicting this tag elevated with the L2 models (i.e., L2-NIKL-o; L2-KLM), when compared to the baseline model (Figure 1).
>
> **Second, the L2-domain specific model may have been adept at identifying prevalent morpheme combination patterns unique to the L2 datasets.** As elaborated in §2, the Korean language has agglutinative nature in which words very frequently encompass multiple morphemes. These morphemes in the datasets are combined with a plus (+) symbol, illustrated in a sentence “나+는 (*na+nun*; I+subjective case marker) 학교+에 (*hak-kyo+ey*; school+oblique case marker) 가+요 (*ka+yo*; go+sentence final marker)” which translates to “I go to school” in English. Given that L2-Korean learners tend to use simpler lexico-grammatical structures in Korean (e.g., Lim et al., 2022; Hwang, 2023), we hypothesized that morpheme combinations in L2 datasets would differ from those in the L1 dataset. **We have thus revised Lines 369-374 as follows:**
>
> “This is in line with previous findings suggesting that domain-specific data can enhance model performance because of the models’ familiarity with patterns found in the target data (e.g., Toutanova et al., 2003; Giménez and Marquez, 2004; Shen et al., 2007). In this study, two patterns emerged as significant. First, the L2 models may effectively detect basic words that are rarely present in the L1 dataset. This is supported by the results that (i) a noun (tagged as NNG) was the most frequently identified POS tag in the datasets (see Table 2) and (ii) the F1 scores predicting this NNG tag elevated with the L2 models (see Figure 1). Second, the L2 models may excel in discerning common morpheme combinations that more frequently appeared in the L2 datasets.  What is particularly evident is the Korean language’s agglutinative nature (see §2.1) where a word can comprise multiple morphemes marked with a plus (+) symbol in the dataset (e.g., 나+는 학교+에 가+요 *na+nun hak-kyo+ey ka+yo* “I go to school”). Given that L2-Korean learners tend to use less sophisticated lexico-grammatical structures (Lim et al., 2022; Hwang, 2023), we postulate a variation in morpheme combinations between native and L2 Korean speakers.”
>
> ---
>
> **5. (Question, Line 215):** it would be useful to expand a bit on what is meant by "same origin" (same speakers?) and why it creates this uncertainty (I assume this is because some L2 errors vary a lot from one speaker to another, so the dataset leaks with respect to the typology of errors? Is that all?).
>
> **Response:** **By "same origin", we meant data originating from comparable settings or conditions.** This includes, but is not limited to, factors such as the data-collection prompt or the nature of the task assigned to participants (in this case, L2 Korean learners). For example, in learner corpus studies, the prompt and input of a task (given to a learner to ask to produce language) may significantly influence the language output (Alexopoulou et al., 2017). **We have elaborated on this term in the revised manuscript as footnote:**
>
> In this context, “same origin” refers to learner data from similar settings or conditions, including factors such as the data-collection prompt and/or the nature of the task assigned to L2 Korean learners for language production. This consideration was informed by previous learner corpus studies that show that the task’s prompt and instructions can markedly influence learner output (see Alexopoulou et al. (2017) for more details).
>
> ---
>
> **6. (Question, Line 326):** is it "consist" (only non-Korean) or "contain" (some non-Korean)? Line 320 seemed to imply "contain"... For instance, if a Korean sentence happens to contain the name of a foreign entity, written in Latin script, is the sentence discarded? This warrants a bit more discussion, at least for more clarity on what has been done, and preferably to explain why it is the right way to do that (or why it is not ideal and only chosen for practical reasons, but it is still OK and does not hurt representativeness because of this or that reason).
>
> **Response:** Our original intention was to filter out sentences from the L2-NIKL dataset that **contained** any non-Korean alphabets, predominantly Chinese characters. This decision was made after observing a subset of L1-Chinese L2-Korean learners who wrote their texts entirely in Chinese, which did not fit our research purposes. However, we now understand that, in certain contexts, the presence of non-Korean characters such as foreign names or technical terms can be relevant even if these terms are written in a non-Korean script. **Given these considerations, we have modified the wording in Line 326 to “contain” and provided a more detailed explanation in the revised manuscript to clarify our decisions and their implications.**
>
> ---
> **7. (Evaluation metrics):**
>
> **7-1. (Shortcoming):** Lack of clarity on the evaluation metrics (see questions below), which renders the numbers quite hard to interpret.
>
> **7-2. (Question, Line 301):** Unclear how "Perfect score" is defined. "comparing the number" is dubious (does not seem to match what is done in the provided reference): only the number, or their boundaries as well? And how is this defined when using Perfect score on the POS task? This is important information that should be clear in the text itself.
>
> **Response:** **“Perfect score” emphasizes the need for perfect alignment between the predicted and gold standard annotations. For a sentence to be perfectly predicted in light of tokenization and POS tagging, every word (or *eojeol* in Korean) in the sentence should match the number of morphemes indicated in the gold standard.** This ensures accuracy in both the tokenization of each morpheme and its subsequent POS tagging (Note that we provided the codes used for this evaluation in the supplemental materials; ‘evaluation.ipynb’).
>
> Given the agglutinative nature of the Korean language in which a single word often contains multiple morphemes, our priority was to ensure that morphological analyzers identified the same number of morphemes as indicated in the gold annotations. **If a word, according to the gold standard, is expected to be segmented into three morphemes but the morphological analyzer divides it into only two, we considered the word mis-parsed even though two morphemes are correctly tagged in alignment with three gold POS annotations.**
>
> While we aimed to explore how previous Korean tokenization/parsing models tackled this issue, we were unable to identify a relevant reference. In this regard, “Perfect score” outlined by Ramen et al. (2022) resonated with our objectives as it similarly addresses the ramifications of imperfect tokenization, especially in languages without spaces (e.g., Thai). The definition in the paper, suggesting that it "reflects the % of times an example is parsed perfectly," closely aligns with our approach.
>
> **To provide clearer insight in the manuscript, we have explicitly elucidated both the motivation behind and the definition of “Perfect score” within our study's context. We have also stressed the significance of counting morphemes before evaluating the F1 scores for tokenization and POS tagging. In addition, we have incorporated examples for readers’ understanding.**
>
> **7-3. (Question, Line 305):** Unclear how the F1 score is defined. Instances=the morphemes? What is the difference between "parsing is perfect" and "tokens align exactly with the golds" (the metric seems tautological)? And the phrasing looks more as an accuracy than an F1 (what do precision and recall mean, with that definition?). For the POS task, is it micro-F1 or macro-F1?
>
> **Response:**
> - In our study, each **"instance"** is referred to as an individual morpheme.
> - What we intended by **"perfect parsing"** was about our model's ability to predict the exact number of morphemes for a token, consistent with the gold standard. Meanwhile, the term **"exact alignment with golds"** refers to the precise match of both the sequence and each morpheme’s tokenization/POS tag with its counterpart in the gold standard. To consider a token as perfectly parsed, it must satisfy both these conditions. (Lines 304-307: “F1 score measures the instances in which predicted tokens/tags align exactly with the golds, but only when the parsing is perfect for the given eojeol.”)
> - Our methodology involves aggregating all true positives, false positives, and so on, across the entire dataset and then calculating precision, recall, and F1, making it a **micro-F1 calculation**.
>
> ---
>
> **8. (Other stylistic, grammatical, formatting issues):**
>
> **8-1. (Adding an outline):**  At the end of §1 it would be good to have some teaser on the outline, so that the readers can identify more directly where to find the information they seek. This can be short, for instance in the style of "After introducing the specificities of L2-Korean (§2), we present in §3...".
>
> **Response:** **We have added an additional paragraph at the end of §1:**
> “This paper is structured as follows: We discuss the importance of morphological analysis in Korean studies and application of morphological analyzers in L2-Korean research (§2). Next, we outline an experiment including datasets and evaluation metrics (§3). We delve into two subsequent experiments: the first involves fine-tuning an L1 model using the full scope of the L2 datasets (§4), while the second focuses on rigorous data cleaning in one of the L2 datasets (§5).  Comprehensive analyses of morpheme tokenization and POS tagging accuracy for both experiments are presented. Finally, we summarize our findings and propose future directions (§6).
>
> **8-2. (Changing titles):** Title of §3: this section is not really about a method, it discusses data, metrics... basically, the experimental conditions. A more usual title for a section like this would be "Experimental setup". Title of §3.2: this section is also about evaluation, so titling it "Model training and evaluation" would be more appropriate. Titles of §4 and §5: any way to have a more explicit title than "Experiment N"? Something that actually summarizes the purpose or conditions of that experiment. That would be helpful to understand the outline, but also in other places of the paper that make references to those experiments, it would certainly help to understand what is referred to and why.
>
> **Response:** We have updated the title of §3 from "Method" to **"Experimental setup"** and the title §3.2 from "Model training" to **"Model training and evaluation"**. We have also revised the titles of §4 and §5 by **adding subtitles** to "Experiment N". For §4, the title is now **"Experiment 1: Maximizing Utility of the L2-NIKL dataset"**, and for §5, **"Experiment 2: Refining and cleaning L2-NIKL Dataset"**. We believe these subtitles to encapsulate the goals of each experiment, which are summarized at the start of the respective experiment sections (§4 Lines 312-314; §5 Lines 413-417).
>
> **8-3: (Line 298):** missing word after "performance"? "model performance measure was undertaken"?
>
> **Response:** We have added "measurement" after the word "performance": "(Line 298) For robust evaluation, model performance measurement was undertaken using the following metrics:"
>
>
> **8-4 (Line 366):** (and same for lines 458, 471, and probably others): It would be better to use "tuned" rather than "trained" in order to better distinguish models trained from scratch from models fine-tuned on that data. A phrasing like "the models trained on L2 data" seems to refer to models which have never seen any L1 data.
>
> **Response:** For improved clarity, we have replaced "trained" with "tuned" in the revised manuscript.
>
>
> **8-5. (Table formatting):** Table 3 (and same for Table 4): The way to read the table is really not obvious (why that many numbers in the same cell), the answer is only given in the caption and part of it is given only implicitly (by means of putting Perfect in parentheses). I would suggest reducing the first column (maybe rotating the labels? or just reducing fontsize?) so that a short second column can be added to indicate the meaning of each line, i.e. adding for instance "F1" / "\Delta_{best}" / "PS" / "\Delta_{best}" vertically just on the right of "L2-NIKL-o Train".
>
> **Response:** To ensure better readability, we have revised both tables in the following way (e.g., Table 3):
>
> | Training | Metric | L2-NIKL-o TOK | L2-NIKL-o POS | L2-KLM TOK | L2-KLM POS |
> |----------|--------|---------------|---------------|------------|------------|
> | L2-NIKL-o | PS | **94.32** | 95.10 | 89.64 | 92.01 |
> | | PS Δbest | - | ↓1.07 | ↓5.46 | ↓3.18 |
> | | F1 | **93.27** | **92.52** | 84.12 | 84.87 |
> | | F1 Δbest | - | - | ↓9.09 | ↓6.03 |
> | L2-KLM | PS | 92.26 | 93.40 | **95.10** | **95.19** |
> | | PS Δbest | ↓2.06|↓2.77| - | - |
> | | F1 | 90.26 | 87.35 | **93.21** | **90.90** |
> | | F1 Δbest | ↓3.01 | ↓5.17 | - | - |
> | L1-GSD | PS | 93.61 | **96.17** | 92.28 | 92.47 |
> | | PS Δbest | ↓0.71 | - | ↓2.82| ↓2.72 |
> | | F1 | 92.36 | 87.52 | 88.98 | 85.37 |
> | | F1 Δbest | ↓0.91 | ↓5.00 | ↓4.23 | ↓5.53 |
>
> ---
>
> **9. (Punctuation/Capitalization/Grammatical issues)**
>
> **Response:** We thank the reviewer for their keen eyes on these matters.
>
> - Footnote 9: issues with parentheses and commas. After "S": missing comma + remove parenthesis. Before "SN": missing comma
> (Revised sentence in Footnote 9: “We excluded tags beginning with 'S', such as SF (sentence-final punctuation marks, e.g., periods), SN (numbers), and SL (foreign characters), as they are not deemed critical for this study.”
> - Table 3's caption (and same for Table 4's caption): missing final dot
> - Line 378 (and same for line 462): "Perfect" with capital P would render the sentence more legible.
> - Line 442: "presents shows" --> "shows"
>
> ---
> **10. References (to be included)**
>
> The following studies are mentioned in the Rebuttal stage but were not included in the submitted manuscript:
>
> - Alexopoulou, T., Michel, M., Murakami, A., & Meurers, D. (2017). Task effects on linguistic complexity and accuracy: A large‐scale learner corpus analysis employing natural language processing techniques. *Language Learning*, 67(S1), 180-208
>
> -  Joshi, P., Santy, S., Budhiraja, A., Bali, K., & Choudhury, M. (2020). The state and fate of linguistic diversity and inclusion in the NLP world. *arXiv preprint arXiv:2004.09095*

---

### Meta-Review · Area_Chair_9YSe · 2023-09-06

**Recommendation:** 2

**Metareview:**

This article describes some interesting work on the quality of morpho-syntactic analysis of learner sentences in Korean, a new task that raises many interesting questions. The reviewers all emphasised the quality and level of detail of the analyses. Many suggestions were made by the reviewers to improve the quality of the presentation. Given the number of modifications requested, I think a second round of reviews is necessary

---

### Decision · Program_Chairs · 2023-10-07

**Decision:**

Accept-Findings

**Comment:**

This article describes some interesting work on the quality of morpho-syntactic analysis of learner sentences in Korean, a new task that raises many interesting questions. The reviewers all emphasised the quality and level of detail of the analyses. Many suggestions were made by the reviewers to improve the quality of the presentation. Given the number of modifications requested, I think a second round of reviews is necessary